# Spin Cross-Over (SCO) Anionic Fe(II) Complexes Based on the Tripodal Ligand Tris(2-pyridyl)ethoxymethane

**Emmelyne Cuza [1], Samia Benmansour [2]**, **Nathalie Cosquer [1]**, **Françoise Conan [1]**, **Sébastien Pillet [3], Carlos J. Gómez-García [2]** and **Smail Triki [1,\***]

[1]  Univ Brest, CNRS, CEMCA, 6 Avenue Le Gorgeu, C.S, 93837–29238 Brest CEDEX 3, France; Emmelyne.Cuza@univ-brest.fr (E.C.); nathalie.cosquer@univ-brest.fr (N.C.); Francoise.Conan@univ-brest.fr (F.C.)
[2]  Instituto de Ciencia Molecular (ICMol), Departamento de Química Inorgánica, Universidad de Valencia, C/Catedrático José Beltrán 2, 46980 Paterna, Spain; sam.ben@uv.es (S.B.); Carlos.Gomez@uv.es (C.J.G.-G.)
[3]  Laboratoire de Cristallographie, Résonance Magnétique et Modélisations, UMR 7036, Boulevard des Aiguillettes, BP239 54506 Vandoeuvre-les-Nancy, France; sebastien.pillet@univ-lorraine.fr
\*   Correspondence: smail.triki@univ-brest.fr; Tel.: +33-298-016-146

**Abstract:** Reactions of Fe(II) with the tripodal chelating ligand 1,1,1-tris(2-pyridyl)ethoxymethane ($py_3$C-OEt) and $(NCE)^-$ co-ligands (E = S, Se, $BH_3$) give a series of mononuclear complexes formulated as $[Fe(py_3C\text{-}OEt)_2][Fe(py_3C\text{-}OEt)(NCE)_3]_2 \cdot 2CH_3CN$, with E = S (**1**) and $BH_3$ (**2**). These compounds are the first Fe(II) spin cross-over (SCO) complexes based on the tripodal ligand tris(2-pyridyl)ethoxymethane and on the versatile co-ligands $(NCS)^-$ and $(NCBH_3)^-$. The crystal structure reveals discrete monomeric isomorph structures formed by a cationic $[Fe(py_3C\text{-}OEt)_2]^{2+}$ complex and by two equivalent anionic $[Fe(py_3C\text{-}OEt)(NCE)_3]^-$ complexes. In the cations the Fe(II) is facially coordinated by two $py_3$C-OEt tripodal ligands whereas in the anion the three nitrogen atoms of the tripodal ligand are facially coordinated and the N-donor atoms of the three $(NCE)^-$ co-ligands occupy the remaining three positions to complete the distorted octahedral environment of the Fe(II) centre. The magnetic studies show the presence of gradual SCO for both complexes: A one-step transition around 205 K for **1** and a two-step transition for compound **2**, centered around 245 K and 380 K.

**Keywords:** tripodal ligands; iron complex; spin cross-over; infrared spectroscopy; magnetic properties

## 1. Introduction

In the last decade, spin cross-over (SCO) complexes are undoubtedly the most studied molecular systems among switchable materials thanks to their several potential applications, in particular, for the development of new generation electronic devices such as memories, molecular sensors and displays [1–10]. These complexes can be reversibly switched from high-spin (*HS*) state to the low-spin (*LS*) state by external physical stimuli such as temperature, pressure, or light irradiation. Even if this spin change can occur in general for some complexes based on transition metal ions with $d^4$-$d^7$ electronic configurations, the Fe(II) octahedral complexes, with $d^6$ electronic configuration, remain, by far, the most investigated systems during recent decades [11–17]. While almost all the SCO materials reported up today are either cationic or neutral [1–17], the number of anionic complexes exhibiting the SCO phenomenon is relatively limited compared to the several hundreds of examples of neutral and cationic complexes reported every year. To the best of our knowledge, all the anionic SCO complexes are based on Fe(III) [18–23] or Fe(II) [24–30]. For those based on Fe(II), three different

systems have been studied. The first system, reported in 2003 [24], concerns a series of Fe(II) complexes of chemical formula [$Fe^{II}H_3L^{Me}$][$Fe^{II}L^{Me}$]X ($X^- = ClO_4^-$, $BF_4^-$, $PF_6^-$, $AsF_6^-$, $SbF_6^-$), based on the hexadentate neutral ligand $H_3L^{Me}$ (tris[2-(((2-methylimidazol-4-yl)methylidene)amino)ethyl]amine) and on the deprotonated ($L^{Me})^{3-}$ anionic form forming a supramolecular 2D-array composed by [$Fe^{II}H_3L^{Me}]^{2+}$ cations and [$Fe^{II}L^{Me}]^-$ anions, both exhibiting SCO. This system can be viewed as interacting anionic and cationic Fe(II) SCO species in a two-dimensional structure. The second example, reported more recently [25], concerns a trinuclear complex [$Fe^{II}_3(\mu\text{-}L)_6(H_2O)_6]^{6-}$, based on the bridging anionic ligand 4-(1,2,4-triazol-4-yl)ethanedisulfonate) (L), exhibiting a SCO transition above room temperature with a large hysteresis loop (>85 K). The first example of the third system was reported in 2012 by T. Ishida et al. [26] as the first SCO example based on the rigid tetrakis(2-pyridyl)methane (py$_4$C) ligand (see Scheme 1a, R = py). In this example, the Fe(II) ion in the mononuclear anionic complex, [Fe(py$_4$C)(NCS)$_3$]$^-$, is surrounded by the py$_4$C tripodal ligand acting as tridentate through the nitrogen atoms of three pyridyl groups and by three thiocyanate anions acting as terminal co-ligands (see Scheme 1b, R = py). As a result of this work and thanks to the chemical flexibility of the tripodal py$_3$C-R ligand that allows chemical modifications by changing the R groups (see Scheme 1a), the same group has reported in 2015 the two first SCO anionic complexes with [Me$_4$N]$^+$ as counter-ion: [Me$_4$N][Fe(py$_3$C-OH)(NCS)$_3$]·(1-PrOH)$_{0.5}$·(H$_2$O)$_2$ [27] and [Me$_4$N][Fe(py$_3$C-Me)(NCS)$_3$]·(H$_2$O) [28], exhibiting gradual SCO at 282 and 330 K, respectively. After annealing at 400 K, the first salt led to the new phase: [Me$_4$N][Fe(py$_3$C-OH)(NCS)$_3$]·(H$_2$O) that presents an abrupt transition with a thermal hysteretic behavior and a significant heating-rate dependence [27,28]. With the objective to understand the effect of the counter-ion and of the functional group of the organic ligand, Ishida et al. have completed their series with two other examples: [(Ph)$_4$P][Fe(py$_3$C-Me)(NCS)$_3$] [29] and [Me$_4$N][Fe((py)$_3$C-$n$-C$_{18}$H$_{37}$)(NCS)$_3$] [30], exhibiting similar mononuclear structures and gradual SCO at 290 and 186 K, respectively.

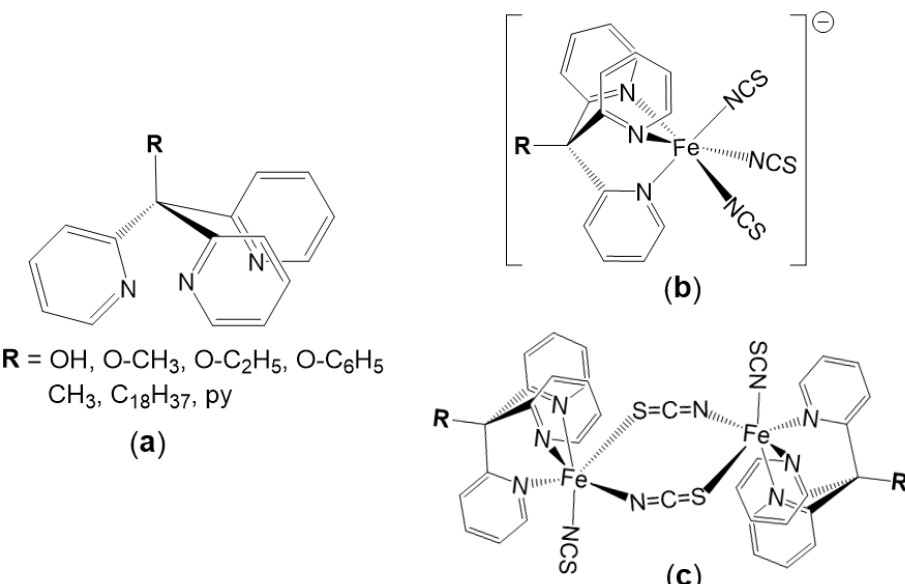

**Scheme 1.** (**a**) Selected functionalized tris(2-pyridyl)methane (py$_3$CR) ligands and examples of tridentate coordination modes observed in anionic (**b**) and neutral (**c**) Fe(II) complexes.

In addition to these SCO salts based on the anionic [Fe(py$_3$C-R)(NCS)$_3$]$^-$ complexes (Scheme 1b), other related systems with different structural features have also been investigated. This concerns the dinuclear [{Fe(py$_3$C-OH)(NCS)($\mu$-NCS)}$_2$]·(PrOH)$_2$ complex reported in 2015 by T. Ishida as the first SCO complex with a FeN$_5$S coordination sphere, in which two [Fe(py$_3$COH)(NCS)] units are doubly bridged by two end-to-end (NCS)$^-$ co-ligands [31] (Scheme 1c, R = OH). More recently, some of us have reported a second dinuclear example, [{Fe(py$_3$C-OC$_6$H$_5$)(NCS)($\mu$-NCS)}$_2$], based on

a similar tripodal ligand (see Scheme 1c, R = $OC_6H_5$), with a double μ-κN:κS-SCN bridge in a head-to-tail configuration [32]. However, the magnetic studies of this compound revealed the presence of ferromagnetic interactions ($J_{FeFe}$ = +1.08 cm$^{-1}$) and the absence of any SCO transition. These two uncommon dinuclear complexes with similar molecular structures exhibiting very different magnetic properties allowed us to perform experimental and theoretical magneto-structural studies that revealed that the large $N_{py}$-Fe$^{II}$ distances and bent N-bound terminal κN-SCN ligands favour the *HS* state of the Fe$^{II}$ ions, while short $N_{tripodal}$-Fe$^{II}$ distances and almost linear Fe-N-C angles favour the *LS* state, allowing the SCO behaviour [32]. At the same time, we have enriched these uncommon polynuclear systems with a new Fe(II) 1-D coordination polymer, [{Fe(py$_3$-OMe)(NCS)(μ-NCS)}$_n$], based on a similar tripodal ligand (see Scheme 1a: R = OMe) and thiocyanato ligand acting as a μ-κN: κS-SCN single bridge in a head-to-tail configuration, for which the magnetic studies showed an abrupt and complete *HS/LS* transition at 199 K [33].

In view of the above observations and in order to further investigate the effect of the functional group of the tripodal ligand and that of the anionic co-ligand, on the structural features and on the SCO characteristics (e.g., transition temperatures, cooperativity and presence of the LIESST effect), we have developed an extensive synthetic work including: (i) Several functionalizations of the py$_3$C-R tripodal ligand by varying the size and the flexibility of the R functional group and (ii) other different anionic co-ligands such as (NCSe)$^-$ and (NCBH$_3$)$^-$ or more sophisticated ones such as cyanocarbanion units involving several coordinating nitrile groups, and for which the geometry allows them to act as either terminal or multi-bridging co-ligands. In the present work, we report the syntheses, structural characterization, variable temperature infrared spectroscopy and magnetic properties of the two isomorph compounds: [Fe(py$_3$C-OEt)$_2$][Fe(py$_3$C-OEt)(NCE)$_3$]$_2$·2CH$_3$CN, with E = S (**1**) and BH$_3$ (**2**) that are the first Fe(II) spin SCO systems involving the tripodal ligand tris(2-pyridyl)ethoxymethane (py$_3$C-OEt) and (NCS)$^-$ and (NCBH$_3$)$^-$ as versatile co-ligands.

## 2. Results and Discussion

### 2.1. Syntheses

The synthesis of tris(pyridin-2-yl)ethoxymethane (py$_3$C-OEt) was carried out in two different steps, following previous published procedures [34,35] with slight modifications: (i) The reaction of one equivalent of 2-bromopyridine in the presence of n-BuLi at −80 °C in THF led to 2-Pyridyllithium. The resulting solution was then treated with one equivalent of 2,2′-dipyridyl ketone leading to a white powder which was characterized as tris(pyridin-2-yl)methanol (py$_3$COH) after purification with a chemical yield of 88%, significantly higher than that reported in references 34–35 (See detailed synthesis and Figures S1–S3 in SI); (ii) the second step consists of a reaction in dimethylformamide under N$_2$ atmosphere at 0 °C of tris(pyridin-2-yl)ethoxymethane in the presence of sodium hydride and iodoethane, leading to a white powder which was characterized as tris(pyridin-2-yl)ethoxymethane (py$_3$C-OEt) with a chemical yield of 85% (See detailed synthesis and Figures S4–S6). Compounds **1** and **2** were obtained as orange and red single crystals, respectively, by mixing a solution of the NCE$^-$ co-ligand E = S (**1**) and BH$_3$ (**2**) with a solution of FeCl$_2$ and tris(pyridin-2-yl)ethoxymethane at −32 °C.

### 2.2. Description of the Structure

According to the spin state derived from the colour at room temperature and to the magnetic properties detailed below, for the two compounds **1** and **2**, the crystal structure of [Fe(py$_3$C-OEt)$_2$][Fe(py$_3$C-OEt)(NCS)$_3$]$_2$·2CH$_3$CN (**1**) was determined at 293 K and 100 K, while the structure of [Fe(py$_3$C-OEt)$_2$][Fe(py$_3$C-OEt)(NCBH$_3$)$_3$]$_2$·2CH$_3$CN (**2**) was determined at 200 K. The unit cell parameters, crystal and refinement data, and the pertinent bond distances and angles are summarized in Tables 1 and 2, respectively, for both compounds.

**Table 1.** Crystal data and structure refinement of complexes [Fe(py₃C-OEt)₂][Fe(py₃C-OEt)(NCE)₃]₂·2CH₃CN (E = S (**1**), BH₃ (**2**)).

|  | **1** | | **2** |
|---|---|---|---|
| Temperature/K | 293 | 100 | 200 |
| Color | Orange | Red | Red |
| Formula | $C_{82}H_{74}Fe_3N_{20}O_4S_6$ | | $C_{82}H_{92}Fe_3N_{20}O_4B_6$ |
| F. Wt. | 1763.52 | | 1654.16 |
| Space group | *P*-1 | | *P*-1 |
| Crystal system | Triclinic | | Triclinic |
| $A$/Å | 11.6683(5) | 11.432(5) | 11.6827(8) |
| $B$/Å | 11.9026(7) | 11.829(5) | 12.0204(10) |
| $C$/Å | 17.1711(9) | 16.857(5) | 16.9162(11) |
| $A$/° | 78.192(5) | 78.072(5) | 78.389(6) |
| $B$/° | 88.279(4) | 88.037(5) | 87.805(6) |
| $\Gamma$/° | 66.544(5) | 65.879(5) | 65.767(7) |
| $V$/Å³ | 2137.9(2) | 2032.4(14) | 2119.3(3) |
| Z | 1 | 1 | 1 |
| $\rho_{calc}$/g cm⁻³ | 1.370 | 1.441 | 1.296 |
| $2\theta$ range (deg) | 6.834–58.856 | 6.527–58.824 | 6.508–58.484 |
| Total reflections | 18,918 | 26,672 | 15,809 |
| Unique reflections/$R_{int}$ | 9830/0.0458 | 9534/0.0722 | 9657/0.0701 |
| Data with $I > 2\sigma(I)$ | 5988 | 6916 | 5051 |
| $N_{var}$ | 523 | 523 | 550 |
| $R_1$ [a] on $I > 2\sigma(I)$/$wR_2$ [b] (all) | 0.0692/0.2053 | 0.0605/0.1752 | 0.0823/0.2752 |
| GooF [c] on F² | 1.028 | 1.066 | 0.997 |
| $\Delta\rho_{max}$ (eÅ⁻³)/$\Delta\rho_{min}$ (eÅ⁻³) | 0.892/−0.703 | 2.110/−1.047 | 1.177/−1.490 |

[a] $R_1 = \Sigma|Fo\text{-}Fc|/Fo)$; [b] $wR_2 = [\Sigma((\omega(Fo^2\text{-}Fc^2))^2/(\omega Fo^2)^2)]^{1/2}$; [c] $GooF = [(\Sigma(\omega(Fo^2\text{-}Fc^2))^2/(N_{obs}\text{-}N_{var})]^{1/2}$.

**Table 2.** Fe-N bond lengths (Å) and $\Sigma$ and $\theta$ distortion parameters (°) for compounds **1** and **2**.

|  | **1** | | **2** |
|---|---|---|---|
|  | **293 K** | **100 K** | **200 K** |
| Fe1-N1 | 2.083(4) | 1.949(3) | 1.944(4) |
| Fe1-N2 | 2.086(4) | 1.956(3) | 1.954(4) |
| Fe1-N3 | 2.110(4) | 1.965(3) | 1.943(5) |
| Fe1-N4 | 2.237(3) | 1.987(3) | 1.978(4) |
| Fe1-N5 | 2.180(3) | 1.948(3) | 1.943(4) |
| Fe1-N6 | 2.191(3) | 1.952(3) | 1.945(4) |
| $<d_{(Fe\text{-}N)}>$ | 2.148(4) | 1.959(3) | 1.951(5) |
| $\Sigma$ | 62.2 | 15.4 | 16.2 |
| $\theta$ | 111.0 | 40.5 | 44.6 |
| Fe2-N7 | 2.003(3) | 1.989(3) | 1.998(4) |
| Fe2-N8 | 1.962(3) | 1.960(3) | 1.947(4) |
| Fe2-N9 | 1.957(3) | 1.954(3) | 1.963(4) |
| $<d_{(Fe\text{-}N)}>$ | 1.974(3) | 1.968(3) | 1.969(4) |
| $\Sigma$ | 18.4 | 17.4 | 19.7 |
| $\theta$ | 43.4 | 43.4 | 46.0 |

$\Sigma$ is the sum of the deviation from 90° of the 12 *cis* angles of the FeN₆ octahedron; $\Theta$ is the sum of the deviation from 60° of the 24 trigonal angles of the projection of the FeN6 octahedron onto its trigonal faces [36].

The crystallographic structural parameters (chemical formula, space group, cell parameters) clearly reveal that both compounds, [Fe(py₃C-OEt)₂][Fe(py₃C-OEt)(NCE)₃]₂·2CH₃CN (E = S (**1**) and

BH$_3$ (**2**)) are isomorph and, therefore, we can expect that main structural differences should arise from the slight differences of their chemical formulae ((NCS)$^-$ for **1** and (NCBH$_3$)$^-$ for **2**). The asymmetric unit of both structures contains one [Fe(py$_3$C-OEt)$_2$]$^{2+}$ cation (Fe2 ion) centred on a crystallographic inversion centre, one [Fe(py$_3$C-OEt)(NCE)$_3$]$^-$ anion (Fe1 ion) and an acetonitrile solvent molecule, both located on general positions (Figure 1 and Figure S7). The Fe(II) ions in both complex units are hexa-coordinated by six N atoms: in the [Fe(py$_3$C-OEt)$_2$]$^{2+}$ cations, the Fe(II) ion (Fe2) is facially coordinated by two crystallographically equivalent (py$_3$C-OEt) tripodal ligands, through the six nitrogen atoms arising from the three pyridyl groups of each tridentate ligand (N7, N8, N9, and N7$^{(i)}$, N8$^{(i)}$, N9$^{(i)}$), while the Fe(II) ion (Fe1) of the anionic complex, [Fe((py$_3$C-OEt)(NCS)$_3$)]$^-$, is surrounded by the three nitrogen atoms (N4, N5, N6) arising from the three pyridyl groups of the tripodal ligand and from the three (NCE)$^-$ anions acting as terminal κN-SCN terminal co-ligands (N1, N2, N3) to complete the distorted octahedral environment of the iron(II) ion (Figure 1). Examination of the Fe-N bond distances and N-Fe-N bonds angles (see below) allows us to describe the metal environment as distorted FeN$_6$ octahedrons in both anionic and cationic complexes in the two compounds (**1** and **2**). The three Fe2-N bond distances (Table 2) in the [Fe(py$_3$C-OEt)$_2$]$^{2+}$ cationic units observed for compounds **1** and **2** (Fe2-N7, Fe2-N8 and Fe2-N9) are quite similar and did not show any significant differences with temperature as indicated by the averaged <d$_{(Fe2-N)}$> values depicted in Table 2 (<Fe2-N>:1.974(3) and 1.968(3) Å for **1** at 293 and 100 K, respectively, and 1.969(4) Å at 200 K for **2**). This low distortion of the coordination sphere with respect to the O$_h$ symmetry in compound **1** (293 and 100 K) and **2** (200 K) is clearly confirmed by almost similar values of their corresponding angular distortion parameter (Σ) [36] depicted in Table 2 (18.4 and 17.4° for **1** at 293 and 100 K, respectively, and 19.7° for **2**). In contrast, the six Fe1-N bond distances in the anionic complex of compound **1** at 293 K show significant differences (Table 2). Thus, the three Fe1-N distances from the N atoms (N1, N2 and N3) of the terminal κN-SCN ligands (2.083(4), 2.086(4) and 2.110(4) Å) are shorter than the bond distances of the nitrogen atoms (N4, N5 and N6) of the tridentate rigid tripodal ligand (2.237(3), 2.180(3), 2.191(3) Å). This deformation of the coordination sphere is further confirmed by the bond angles that deviate considerably from the ideal values (the *cis* angles range from 79.16° to 94.90° whereas the trans angles are in the range 167.52° to 174.87°) and by the relatively high values of the distortion parameter (Σ = 62.2°) [36]. At low temperature (100 K), this strong distortion is clearly reduced as shown by the six Fe1-N almost equivalent bond distances and by a much lower distortion parameter (Σ = 15.4°). Similarly, the anionic complex [Fe(py$_3$C-OEt)(NCBH$_3$)$_3$]$^-$ part of compound **2** at 200 K, displays equivalent Fe1-N bond distances (1.943–1.978 Å) and low distortion parameter (Σ = 16.2°) which are almost similar to those observed for the cationic units [Fe(py$_3$C-OEt)$_2$]$^{2+}$ of **1** and **2**. (Σ = 17.4–19.7°).

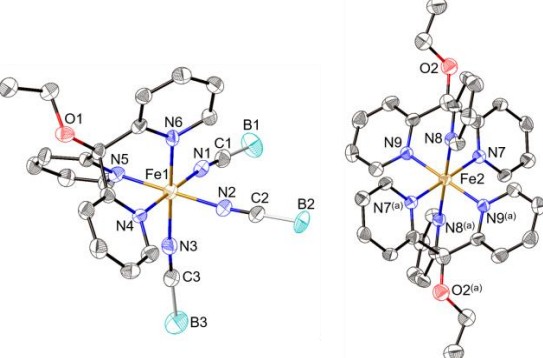

**Figure 1.** View of the anionic [Fe(py$_3$C-OEt)(NCBH$_3$)$_3$]$^-$ and the dication [Fe(py$_3$C-OEt)$_2$]$^{2+}$ complexes in [Fe(py$_3$C-OEt)$_2$][Fe(py$_3$C-OEt)(NCBH$_3$)$_3$]$_2$·2CH$_3$CN (**2**) showing the atom labelling scheme and the coordination environment of the two Fe(II) ions (Fe1 and Fe2). Similar structure for [Fe(py$_3$C-OEt)$_2$][Fe(py$_3$C-OEt)(NCS)$_3$]$_2$·2CH$_3$CN (**1**). Codes of equivalent positions: (a) = −x, −y, −z.

Finally, it is worthy to note that the different coordination spheres described above can be highlighted according to the values of the Fe-N distances and of the distortion parameters which are known to be highly sensitive to the spin state of metal ion. Indeed, the spin state (high-spin (*HS*) or low-spin (*LS*)) of each Fe(II) ion can be assigned according to the average value of the Fe-N bond distances (<Fe-N>$_{LS}$ ≈ 2.0 Å; <Fe-N>$_{HS}$ ≈ 2.2 Å) and to the value of the distortion parameters since high degree of distortion of the FeN$_6$ octahedron is indicative of *HS* configuration [11–17,36]. Ultimately, the <Fe-N> values (1.974(3) and 1.968(3) Å for **1** at 293 and 100 K, respectively, and 1.969(4) Å for **2** at 200 K) and the relatively low Σ parameters (18.4° for **1** at 293 K, 17.4° for **1** at 100 K and 19.7° for **2**) observed for the cationic complexes [Fe(py$_3$C-OEt)$_2$]$^{2+}$ in both compounds are characteristic of the *LS* state. In contrast, the much more distorted coordination sphere observed for the anionic complex in **1** at 293 K (<Fe-N> = 2.148(4) Å, Σ = 62.2°, θ = 111.0°), associated with its lower distortions (Σ = 15.4°, θ = 40.5°) and to significant contraction (<Fe-N> = 1.959(3) Å) at 100 K, reveal the presence of a SCO in compound **1** (see discussion below).

### 2.3. Variable Temperature Magnetic Properties and Infrared Spectroscopy

In line with the structural conclusions above mentioned, variable temperature susceptibility measurements were performed in the temperature range 2–400 K for **1** and 2–500 K for **2**. The thermal variation of the product of the molar magnetic susceptibility times the temperature ($\chi_m T$) are shown in Figure 2 for both complexes.

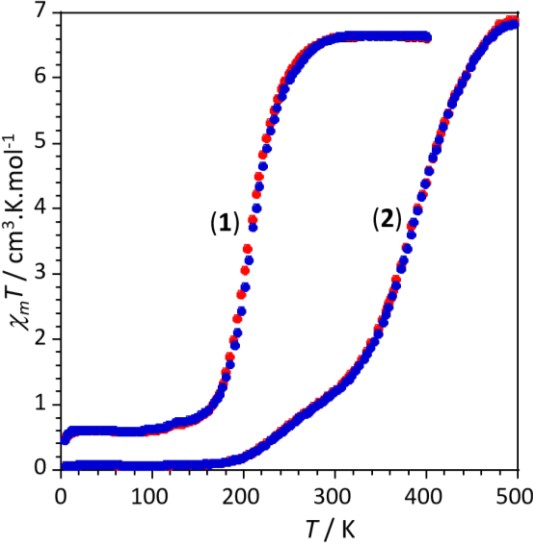

**Figure 2.** Thermal variation of the $\chi_m T$ product for complexes **1** and **2** in the cooling (blue dots) and warming (red dots) scans.

The $\chi_m T$ values per formula (two Fe(II) ions) at 400 K for compound **1** (≈6.7 cm$^3$ K mol$^{-1}$) and at 500 K for compound **2** (≈6.9 cm$^3$ K mol$^{-1}$) are the expected spin only values for two isolated metal ions with $S = 2$ and g ≈ 2.1 and indicate the presence of two magnetically isolated Fe(II) ions in the *HS* state ($S = 2$) [11–17]. Upon cooling, the $\chi_m T$ value of **1** remains almost constant down to approximately 245 K where it shows a quite abrupt decrease reaching a plateau of ca. 0.65 cm$^3$ K mol$^{-1}$ below 150 K, corresponding to a residual *HS* fraction of ca. 10%. This behaviour indicates the presence of an incomplete *HS* to *LS* transition at $T_{1/2} = 205$ K, as also revealed by the thermochromism observed in the single crystals (orange at 293 K and red below 150 K). The $\chi_m T$ product of **2** shows an abrupt decrease from 500 K until approximately 360 K and then decreases in a much more gradual way reaching a value of 0.04 cm$^3$ K mol$^{-1}$ at ca. 180 K that remains constant below this temperature, indicating the presence of a complete and gradual *HS* to *LS* two-step-like transition centred at around 245 K and 380 K. For both complexes the magnetic properties were measured in both cooling and warming modes, but

no significant hysteretic effects were detected. We have also performed photomagnetic studies of both samples by irradiating the samples with a green laser at 10 K for several hours without detecting any noticeable increase of the magnetic moment in both samples.

To confirm the Fe(II) spin state at high and low temperatures, the presence of the incomplete *HS* to *LS* transition for **1** with a residual *HS* fraction of ca. 10% in the *LS* region (see Figure 2) and the presence of an almost complete spin transition for **2**, we have performed infrared spectroscopy in the 2150–2200 cm$^{-1}$ range corresponding to the fundamental stretching vibration of the NCS and NCBH$_3$ moieties. It is well known that the intensity of these stretching vibrations is very sensitive to the spin state of the metal ion [37–43]. According to the thermal dependences of the $\chi_m T$ products depicted in Figure 2 for the two complexes, we have studied the NCS stretching vibrations for **1** at 300 and 150 K, and the NCBH$_3$ vibrations at 500 and 170 K for **2**. The final infrared spectra in the region of the C-N frequency for the *HS* and *LS* of each compound are depicted in Figure 3.

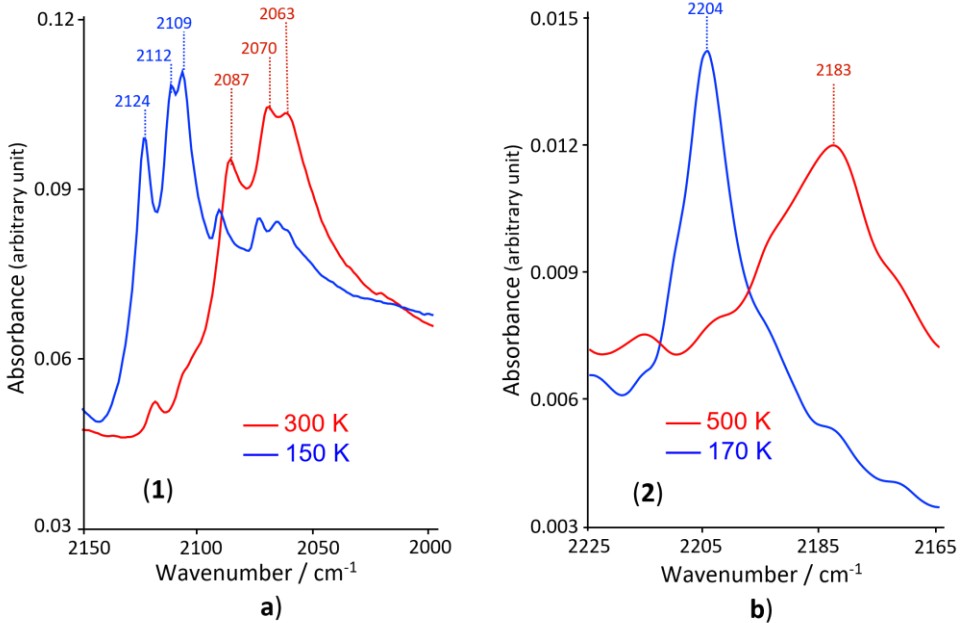

**Figure 3.** Infrared spectra for compounds **1** (**a**) and **2** (**b**) in the 2150–2200 cm$^{-1}$ region, showing the principal bands that are temperature sensitive.

Both complexes, show a clear evolution of the ν(CN) modes with temperature, in agreement with the presence of the SCO: At 300 K, complex **1** presents three strong vibration modes around 2063, 2070 and 2087 cm$^{-1}$(see Figure 3a, red band), while at 150 K, corresponding to the *LS* region (see Figure 2), three strong bands, characteristic of the *LS* state, appear at higher wavenumbers (2109, 2112 and 2124 cm$^{-1}$), in addition to the persistence of the three bands observed at 300 K, but with lower intensities (Figure 3a, blue band). This spectrum confirms the predominance of the *LS* state at 150 K and the persistence at low temperature of the three weak *HS* characteristic bands, confirms the presence of a significant *HS* fraction at 150 K, in agreement with the magnetic studies.

For complex **2**, the infrared spectrum at 500 K shows a large strong band around 2183 cm$^{-1}$ characteristic of the *HS* spin state, in agreement with the magnetic studies (see Figure 3b, red band), while at 170 K a strong band appears around 2204 cm$^{-1}$ (see Figure 3b, blue band) which should be attributed to the *LS* state as revealed by the magnetic studies. The temperature dependence of the infrared spectrum of complex **2** does not show significant persistence of the strong characteristic bands of the *HS* state at low temperature, confirming the complete nature of the *HS* to *LS* transition in compound **2**.

### 2.4. Magneto-Spectroscopic and Structural Relationships

Based on the transition temperatures derived from the magnetic studies above and on the temperature regions of the SCO transition in **1** and **2**, the crystal structure of **1** has been determined at 293 and 100 K, corresponding, respectively to the *HS* and *LS* states; while the crystal structure of **2** has been only studied at 200 K to characterize the *LS* state since the *HS* state, expected above 500 K, could not be reached by our single crystal X-ray equipment. However, this lack of high temperature structural data is compensated by the infrared studies performed in the temperature range 100–500 K since it is well known that the infrared spectroscopy is a very useful tool for the characterization of the *HS* and *LS* states [37–43]. Therefore, the average value of the Fe-L distances (Fe-N) and the distortion parameter ($\Sigma$) derived from the X-ray single crystal structural data [11–17,36], as well as the thermal evolution of the infrared vibration bands, which are highly sensitive to the Fe(II) spin state, will be used in this section to confirm the spin state on the Fe(II) centres.

Table 1 lists the average values of the Fe-N bond lengths and that of the $\Sigma$ and $\theta$ distortion parameters for all the Fe(II) centres of the three structures (structure of **1** at 293 and 100 K, and structure of **2** at 200 K). Similarly, it is noteworthy that each structure involves two iron(II) centres, the first one (Fe1), from the anionic complex $[Fe(py_3C\text{-}OEt)(NCE)_3]^-$ (E = S (**1**), $BH_3$ (**2**)), and the second one (Fe2) from the cationic complex $[Fe(py_3C\text{-}OEt)_2]^{2+}$. The average values of the Fe-N ($<d_{(Fe\text{-}N)}>$) bonds of the coordination sphere of the cation (1.973(2) Å and 1.963(2) Å for **1** at 100 and 300 K, respectively and 1.970(3) Å for **2** at 200 K) as well as the corresponding value of the $[Fe(py_3C\text{-}OEt)(NCS)_3]^-$ anion in **1** at 100 K (1.958(3) Å) are very similar and agree with the *LS* state nature of the Fe2 centres in all cationic complexes and in the anionic unit of **1** at 100 K, as also confirmed by the relatively low value of the $\Sigma$ distortion parameter (in the range 15.1–19.1°) [11–26,29]. Thus, among all the metal centres, only the Fe1 ion from the anionic complex $[Fe(py_3C\text{-}OEt)(NCS)_3]^-$ of **1** at 293 K displays higher average $<d_{(Fe\text{-}N)}>$ value (2.146(3) Å) and more important distortion ($\Sigma$ = 62.2°, $\theta$ = 111.0°) which are in good agreement with the corresponding values observed for the *HS* Fe(II) ion in a $FeN_6$ distorted octahedral environment [11–17,24–33,36] and with the magnetic data.

The crystal structural characterization shows that the two complexes display discrete mononuclear structures which differ only in the three ancillary ligands (Figure 1). The overall crystal packing is assisted by intricate C-H⋯S hydrogen bonds for **1**, which are very similar in compound **2** through C-H⋯B interactions (Figure S8a), together with $\pi$-$\pi$ interactions between the $[Fe(py_3C\text{-}OEt)(NCE)_3]^-$ anionic complexes (Figure S8b). As shown in Figure S9, no significant strong intermolecular interactions are really dominant in the crystal packing, suggesting the absence of any significant cooperative effects as revealed by the gradual switching behaviours and the lack of hysteresis shown in Figure 2. However, the two compounds differ markedly by their overall transition temperatures since compound **1** exhibits a one-step transition centred at 205 K; while complex **2** displays higher transition temperatures with a two-step behaviour at around 245 K and 380 K. This observation led us to examine the Fe-N-CS bond angles in the two complexes. According to one of our recent experimental and theoretical magneto-structural work on SCO systems involving NCS units as ancillary co-ligands, some of us suggested that the bent configuration of the N-bound terminal thiocyanato ligand promotes a weaker ligand field on the Fe(II) ion than the linear one [32,33]. Examination of the Fe-N-CS angles summarized in Table 3 for the three crystal structures, clearly shows that the Fe-N-C(S) angles observed for complex **1** at 293 K present the higher deviation (<Fe-N-C> = 166.7(4)°) with respect to the linearity, with similar trend between the Fe-N-C(S) angles observed for **1** at 100 K (<Fe-N-C> = 171.7(3)°) and the Fe-N-C(B) angles observed for **2** at 200 K (<Fe-N-C> = 174.0(4)°). These structural observations suggest the following classification of the ligand field strength (**1** (*HS*, 293 K) < **1** (*LS*, 100 K) < **2** (*LS*, 200 K)) [32,33], and confirms, as expected, that the ligand field is stronger in complex **2** involving the $NCBH_3^-$ ancillary co-ligand, in agreement with its higher transition temperature derived from the magnetic studies [44].

**Table 3.** Fe-N-C bond angles (°) arising from the bent N-bound terminal κN-ECN coordination mode (E = S (**1**), BH$_3$ (**2**)).

|  | 1 | | 2 |
|---|---|---|---|
|  | 293 K | 100 K | 200 K |
| Fe1-N1-C1 | 165.7(4) | 171.0(3) | 172.5(4) |
| Fe1-N2-C2 | 159.7(4) | 168.5(3) | 174.1(4) |
| Fe1-N3-C3 | 174.7(4) | 175.7(3) | 175.3(4) |

As mentioned above, the infrared spectroscopy has been used in the present work to know more on the thermal evolution of the *HS* and *LS* state of both compounds. According to the chemical nature of the two compounds, their spin can be, in principle, studied using either stretching vibration of NCS and NCBH$_3$ moieties or those of the Fe-N bonds. However, we have decided to focus our efforts on the thermal dependence of the infrared spectra of the stretching vibration of NCS and NCBH$_3$ moieties (2150–2200 cm$^{-1}$ in the case of complexes **1** and **2**). That choice was justified by the fact that each compound is composed of two different Fe(II) complexes: [Fe(py$_3$C-OEt)$_2$]$^{2+}$ cationic complexes involving only *LS* Fe(II) centres, and [Fe(py$_3$C-OEt)(NCS)$_3$]$^-$ (E = S (**1**) and BH$_3$ (**2**)) anionic complexes exhibiting thermal spin transition; this should allow clearer discrimination of the three possible spin state configurations (*HS* or *LS* single spin state, and *HS/LS* mixed spin state). Thus, for each compound, we have recorded the infrared spectra in the vicinity of the SCO transitions from 300 to 150 K for **1** (Figure 4a) and from 500 to 170 K for **2** (Figure 4b).

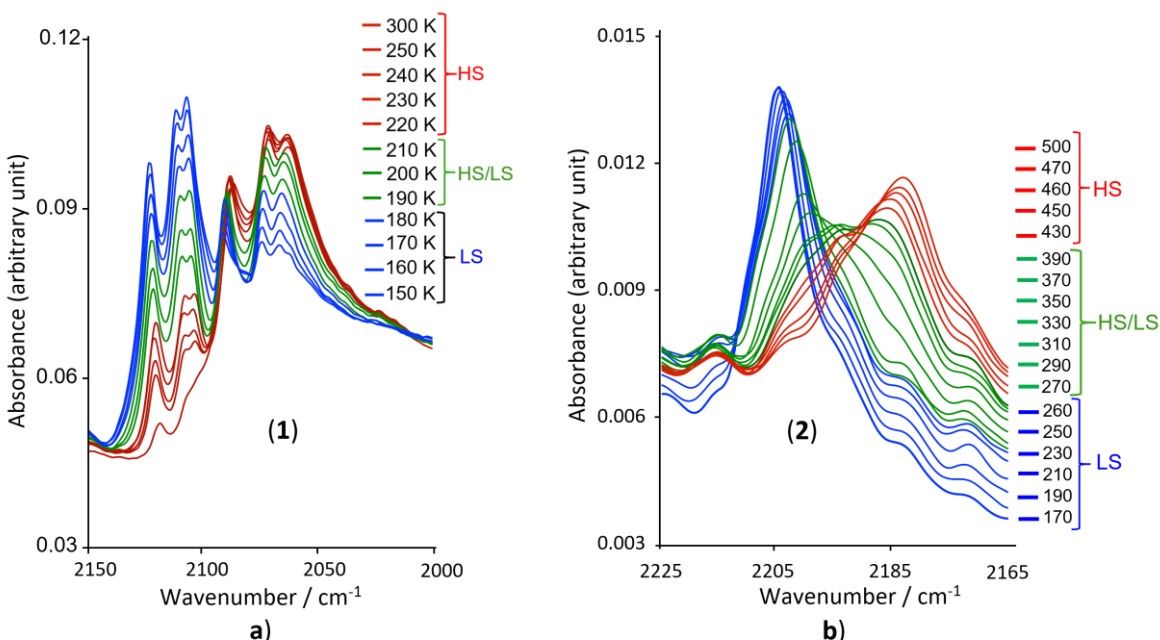

**Figure 4.** Temperature dependences of the infrared spectra in the temperature region 300–150 K, for compound **1** (**a**) and in the temperature region 500–170 K for **2** (**b**).

For **1**, the intensity of the three ν(NCS) vibrational bands observed at 2063, 2070 and at 2087 cm$^{-1}$, characteristic of the *HS* state, gradually decrease with decreasing the temperature from 300 to 150 K. As discussed above, even at 150 K, the three infrared bands persist (Figure 4a), suggesting the presence of a residual Fe(II) *HS* state, in agreement with the magnetic studies. As the intensity decreases, three other bands, characteristic of the *LS* state, appear at higher wavenumbers (2109, 2112 and 2124 cm$^{-1}$) whose intensities gradually increase with decreasing the temperature. Similar trend was observed for the ν(NCB) vibrational bands of compound **2** (Figure 4b); the intensity of characteristic band of the *HS* state, observed at 2183 cm$^{-1}$, decreases with decreasing the temperature from 500 to 170 K, while a new

band, characteristic of the *LS* state, appears at higher wavenumbers (2204 cm$^{-1}$). To better assign each vibrational band to the corresponding spin state and to study the global consistency of the experimental data of the thermal dependence of the infrared spectroscopy in both compounds, we have correlated the results of the thermal dependence of the $\chi_m T$ product derived from magnetic measurements and those of the thermal evolution of the intensity of the infrared bands for both compounds. The data, summarized in Figure 5, show an excellent correlation between the thermal evolution of the $\chi_m T$ product and the intensity of one of the *HS* characteristic bands (2063 cm$^{-1}$ for **1** and 2183 cm$^{-1}$ for **2**), showing a significant change in the temperature region where the thermal spin transition occurs in the magnetic data for both compounds.

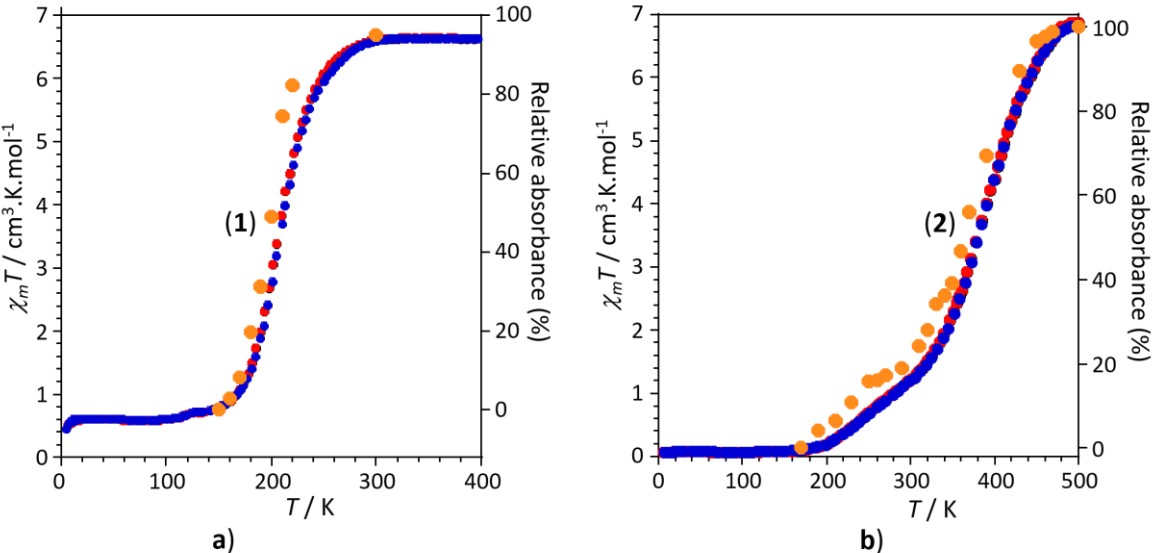

**Figure 5.** (**a**) Temperature dependences of the $\chi_m T$ product and of the relative absorbance of the ν(CN) band observed at 2063 cm$^{-1}$ (•) for compound **1**. (**b**) Temperature dependences of the $\chi_m T$ product and of the relative absorbance of the ν(CN) band observed at 2183 cm$^{-1}$ (•) for compound **2**.

Finally, it is important to note that one of the more intriguing observations in this contribution concerns the SCO behaviour observed for compound **2** that shows two-step transition centred around 245 K and 380 K, while compound **1** exhibits a residual *HS* fraction of ca. 10% below 150 K. This unexpected behaviour was also confirmed by the temperature dependence of the infrared spectra as clearly shown in Figure 5b for which the data derived from the infrared studies (thermal variation of the intensity of the infrared band observed at 2063 cm$^{-1}$) perfectly fits with the two steps for **2** and the residual fraction for **1** revealed by the magnetic data.

Juxtaposition of the two-step behaviour (500–320 K and 320–200 K, see Figure 5b) of compound **2** with that observed for complex **1** in the temperature range of 260–160 K (see transition region in Figure 5a) reveals that the profile of the $\chi_m T$ vs. *T* portion in the range 500–320 K (first step of **2**) is close to that of **1** (see temperature range 260–100 K). Such juxtaposition, clearly depicted in Figure 6a, corresponds to that expected for two isomorph complexes for which the magnetic behaviours should only be differentiated by their transition temperatures, according to their ancillary co-ligands (NCS$^-$ for **1** and NCBH$_3^-$ for **2**) that promotes different ligand field energies. We can therefore conclude that the first step observed for **2** in the range 500–320 K may be related to the isomorph crystal structures described above for both complexes, while the second step may be induced by unexpected structural changes.

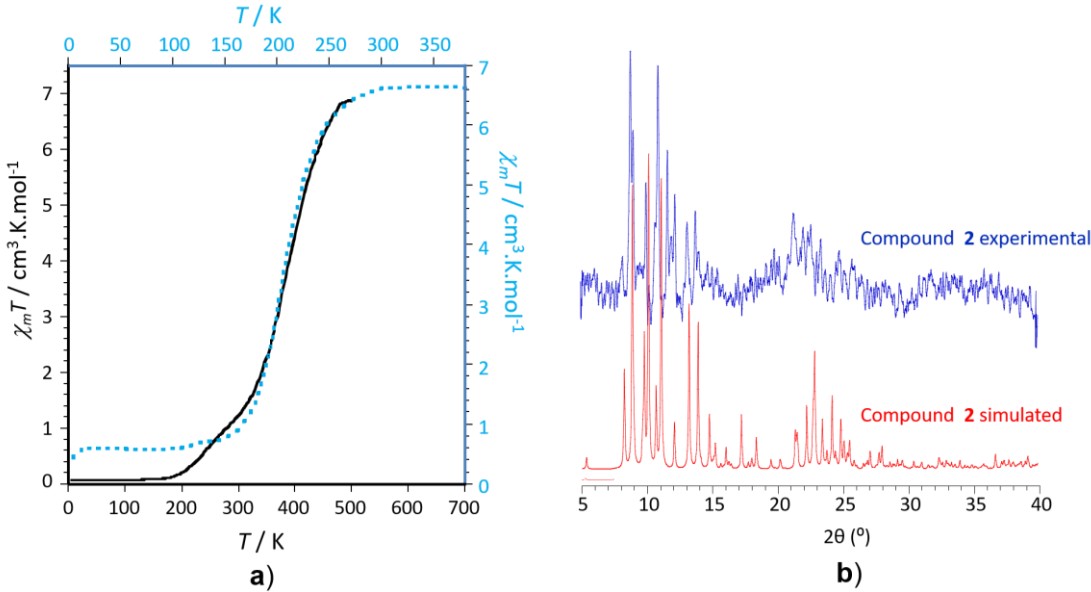

**Figure 6.** (**a**) Temperature dependences of the $\chi_m T$ product of **1** (black solid line) and **2** (blue dashed line) showing similar behaviours above 320 K, when the transition temperatures are not taken into account; (**b**) observed and calculated X-ray powder diffraction patterns for compound **2**.

To shed light on the presence of such second step, we have performed additional X-ray experiments at room temperature near the centre of the second step observed for **2**: (i) The single crystal structure could not be determined correctly at 293 K because of the low diffraction power of the single crystals due to the low crystallinity of the sample. This was confirmed by the powder X-ray diffraction recorded on crushed single crystals which does not give a well-resolved X-ray diffraction pattern (see Figure 6b and Figure S10). The main diffraction peaks identified from the simulated diffraction pattern fits perfectly with the observed ones, while no other strong peaks are observed in the experimental pattern, which confirms first the purity of the powder sample and that no additional crystalline phase coexist, which could have corresponded to the second transition step. On the contrary, the observed diffraction pattern shows strong broad diffuse scattered signal, which is most probably the signature of an amorphous phase, which can be at the origin of to the second step. A different process is demonstrated by the powder X-ray diffraction for compound **1**. In this case, the simulated and measured diffraction patterns match perfectly without any diffuse signal, ruling out the possibility of an additional amorphous phase. In that case, the residual fraction is present in the well crystalline phase, and contributes to the derived single crystal structure. As a matter of fact, the Fe-N and $<d_{(Fe-N)}>$ bond distances are systematically slightly higher for compound **1** at 100 K, by comparison with compound **2** at 200 K (see Table 2).

## 3. Experimental Section

### 3.1. Starting Materials

Organic starting compounds, iron salts, and solvents were purchased from Sigma-Aldrich and Acros Organic. Deuterated solvents were purchased from Cambridge Isotope Laboratories. Dried solvents were prepared by refluxing for one day under nitrogen over appropriate drying agents (sodium and benzophenone for tetrahydrofuran (THF), calcium hydride for methanol, barium oxide under vacuum for dimethylformamide (DMF), sodium for ethanol), others solvents used were dried through drying column. All the ligand syntheses were carried out under nitrogen atmosphere by using a dual manifold vacuum/nitrogen line and standard Schlenk techniques.

### 3.2. Syntheses of Tris(pyridin-2-yl)methanol (py₃C-OH) and Tris(pyridin-2-yl)ethoxymethane (py₃C-OEt)

The syntheses were performed following the previous published procedures [34,35] with slight modifications which induce more significant yields.

#### 3.2.1. Synthesis of Tris(pyridin-2-yl)methanol (py₃C-OH)

A solution containing 2-bromopyridine (6.32 g; 3.8 mL; 40.0 mmol) and 50 mL of distilled THF was cooled to −80 °C under nitrogen atmosphere. Then n-BuLi (20.0 mL, 50.0 mmol) was added dropwise leading to a bright red solution. After stirring for 15 min at −80 °C, a solution of 2,2′-dipyridyl ketone (7.36 g, 40.0 mmol), which was dissolved beforehand in 50 mL of THF, was slowly added to avoid an increase of the temperature. The resulting marine blue solution was then stirred at −80 °C until the coloration became deep purple, and stirred again for 2 h at room temperature (see Scheme S1). The reaction was quenched with 30 mL methanol, and warmed to room temperature. The final solution was filtered to remove the white precipitate and 100 mL of water was added. The aqueous phase was extracted with 200 mL of dichloromethane, dried over magnesium sulfate, and evaporated using a rotary evaporator. The brown oil resulting compound was purified using 5 mL of acetone and left in the freezer overnight at −28 °C, to obtain a white powder (9.26 g, 35.2 mmol, yield 88%). $^1$H NMR (CDCl$_3$, 400 MHz, δ (ppm)) of py$_3$C-OH at 25 °C: 7.20 (3H, CH aromatic, q, 3JH-H = 5 Hz, H3), 7.68 (3H, CH aromatic, td, H4), 7.75 (3H, CH aromatic, q, 3JH-H = 8 Hz, H5), 8.46 (3H, CH aromatic, 3JH-H = 5 Hz, td, H2). $^{13}$C NMR (CDCl$_3$, 75 MHz, δ (ppm)) of py$_3$COH at 25 °C: 81.27 (C7), 121.36 (C3), 122.01 (C5) 135.46 (C4), 146.87 (C2), 161.99 (C6).

#### 3.2.2. Synthesis Tris(pyridin-2-yl)ethoxymethane (py₃C-OEt)

Under Nitrogen atmosphere, in a 50 mL flask was dissolved tris(pyridyl)methanol (400 mg, 1.52 mmol) in 4 mL of distilled *N,N′*-dimethylformamide. Then sodium hydride was quickly and carefully added (180 mg, 7.50 mmol) and the mixture was stirred for 30 min at 0 °C before adding dropwise 1-iodoethane (249 mg, 165 μL, 1.60 mmol). After stirring for 5 h at room temperature, the reaction was quenched with 20 mL of acetone/methanol (1:1) solution (see Scheme S2). Then a volume of 10 mL of water was added before evaporating the organic phase with a rotary evaporator. The crude product was extracted with dichloromethane and purified in a 1:4 mixture of acetone: Pentane, one night in the freezer. The molecule was obtained as a white powder (375 mg, 1.2 mmol, 85%). $^1$H NMR (CDCl3, 400 MHz, δ (ppm)) of py$_3$C-OEt at 25 °C: 1.24 (3H, CH3, t, 3JH-H = 7.2 Hz, H10), 3.37 (2H, CH2, q, 3JH-H= 7.2 Hz, H9), 7.13 (3H, CH aromatic ring, q, 3JH-H = 8 Hz, H3), 7.65 (3H, CH aromatic ring, q, 3JH-H = 8 Hz, H5), 7.73 (3H, CH aromatic ring, d, 3JH-H = 8 Hz, H4), 8.57 (2H, CH aromatic ring, d, 3JH-H = 4.8 Hz, H2). $^{13}$C NMR (CDCl3, 125 MHz, δ (ppm)) of py$_3$C-OEt at 25 °C: 15.56 (C10), 60.67 (C9), 88.11 (C7), 122.03 (C3), 123.61 (C5) 136.16 (C4), 148.63 (C2), 161.84 (C6).

### 3.3. Synthesis of [Fe(py₃C-OEt)₂][Fe(py₃C-OEt)(NCE)₃]₂·2CH₃CN (E = S (1), NCBH₃ (2))

In 5 mL of distilled methanol were dissolved tri(pyridin-2-yl)ethoxymethane (50.0 mg, 0.17 mmol), iron(II) chloride salt (20.0 mg, 0.16 mmol) and a few mg of ascorbic acid. The resulting solution was stirred for 15 min at room temperature and then a solution of acetonitrile (5 mL) containing 0.69 mmol of (C$_2$H$_5$N)(NCE) was added. The resulting solution was stirred for 30 min and then filtered and placed quickly at −32 °C. After a few days, orange and red prismatic single crystals of **1** and **2**, respectively, were recovered. Anal. Calcd. (%) for [Fe(py$_3$C-OEt)$_2$][Fe(py$_3$C-OEt)(NCS)$_3$]$_2$·2CH$_3$CN (C$_{82}$H$_{74}$Fe$_3$N$_{20}$O$_4$S$_6$), **1**: C, 55.8; H, 4.2; N, 15.9; Found (%): C, 55.4; H, 4.0; N, 15.3. IR data (ν/cm$^{-1}$) for the freshly filtered sample: 410w, 423w, 477w, 500w, 513w, 530w, 659m, 726w, 758w, 886w, 1011m, 1086w, 1108m, 1143m, 1205w, 1252w, 1291w, 1389w, 1434m, 1462s, 1593m, 2060s, 2244w, 2871w, 2901w, 2972.11w, 3076w, 3442br. Anal. Calcd. (%) for [Fe(py$_3$C-OEt)$_2$][Fe(py$_3$C-OEt)(NCBH$_3$)$_3$]$_2$·2CH$_3$CN (C$_{82}$H$_{92}$Fe$_3$N$_{20}$O$_4$B$_6$), **2**: C, 59.4; H, 5.6; N; 16.9. Found (%): C, 59.0; H, 5.4; N, 16.3. IR data (ν/cm$^{-1}$) for the freshly filtered sample: 467w, 491w, 481w, 512m, 530m, 558w, 591w, 616w, 684w, 743w, 759s, 774s,

798s, 863m, 871m, 890w, 900w, 946w, 1012 m, 1067w, 1087s, 1109w, 1123s, 1135w, 1142m, 1163s, 1205m, 1248w, 1298m, 1394w, 1439s, 1460s, 1595m, 2168m, 2219w, 2312s 2329m, 2919w, 2982w, 3071w), 3105w.

### 3.4. Characterization of the Materials

Elemental analyses were performed by the "Service Central d'Analyses du CNRS", Gif-sur-Yvette, *France*. Room temperature infrared spectra of ligands and complexes were recorded on a platinum ATR Vertex 70 BRUKER spectrometer in the range 4000–400 cm$^{-1}$. $^1$H et $^{13}$C NMR spectra were recorded using BRUKER DRX 300 MHz, Advance 400 MHz and Advance III HD 500 MHz equipment (University of Brest). All chemical shifts are defined in ppm and determined by using the rightful deuterated solvent as a reference.

### 3.5. Magnetic Measurements

Magnetic susceptibility measurements were carried out in the temperature range 2–400 K for **1** and in the range 2–500 K for **2** with an applied magnetic field of 0.1 T, on polycrystalline samples of both compounds (with masses of 11.634 mg for **1** and 4.065 mg for **2**) with a Quantum Design MPMS-XL-5 SQUID susceptometer (San Diego, CA, USA) using a sample space oven for compound **2** (in the range 300–500 K). Consecutive cooling and warming scans at a scan rate of 2 K min$^{-1}$ showed identical behaviours without any noticeable hysteresis. The susceptibility data were corrected for the sample holders previously measured using the same conditions and for the diamagnetic contributions of the salt as deduced by using Pascal's constant tables [45]. The photomagnetic studies were performed by cooling the samples down to 10 K at a rate of 1 K min$^{-1}$ and then irradiating them with a green Diode Pumped Solid State Laser DPSS-532-20 from Chylas (λ = 532 nm, power = 20 mW) coupled via an optical fibre to the cavity of the SQUID magnetometer.

### 3.6. Crystallographic Data Collection and Refinement

The crystallographic studies were performed at room temperature and at 100 K for **1**, and at 200 K for **2**, using an Oxford Diffraction Xcalibur κ-CCD diffractometer equipped with a graphite monochromated Mo*Kα* radiation (λ = 0.71073 Å). The full sphere data collections were performed using 1.0° ω-scans with different exposure times per frame (225 s for **1** at 293 K, 45 s for **1** at 100 K, and 175 s for **2** at 200 K). Data collection and data reduction were done with the CRYSALIS-CCD and CRYSALIS-RED programs on the full set of data [46]. The crystal structures were solved by direct methods and successive Fourier difference syntheses, and were refined on $F^2$ by weighted anisotropic full-matrix least-square methods [47]. All non-hydrogen atoms were refined anisotropically and the hydrogen atoms were calculated and included as isotropic fixed contributors to $F_c$. All other calculations were performed with standard procedures (*OLEX2*) [48]. Crystal data, structure refinement, and collection parameters are listed in Table 1.

X-ray powder diffraction (XRPD) data were collected for both compounds on polycrystalline samples filled into 0.7 mm glass capillaries that were mounted and aligned on a Empyrean PANalytical powder diffractometer, at 45 kV, 40 mA using Cu Kα radiation (λ = 1.54177 Å). A total of four scans were collected for each sample, at room temperature in the 2θ range of 5–40°. The isostructurality of the samples prepared was confirmed with the experimental and simulated powder X-ray diffractograms (Figure S10).

CCDC-1998534-1998536 contains the supplementary crystallographic data for compounds **1** (293 K and 100 K) and **2** (200 K), respectively. These data can be obtained free of charge from The Cambridge Crystallographic Data Center at www.ccdc.cam.ac.uk/data_request/cif.

## 4. Conclusions

Two novel SCO Fe(II) complexes: [Fe(py$_3$C-OEt)$_2$][Fe(py$_3$C-OEt)(NCE)$_3$]$_2$·2CH$_3$CN, with E = S (**1**), NCBH$_3$ (**2**), based on the tris(2-pyridyl)*ethoxymethane* (py$_3$C-OEt) tripodal ligand and the two ancillary NCS$^-$ and NCBH$_3$$^-$ co-ligands, have been prepared. Complex **1** has been structurally characterized

at room temperature (293 K) and at 100 K, while complex **2** has been only characterized at 200 K. Both compounds display almost similar molecular structures composed by similar cationic complex ($[Fe(py_3C\text{-}OEt)_2]^{2+}$) and anionic complex ($[Fe(py_3C\text{-}OEt)(NCE)_3]^-$, with E = S (**1**), $NCBH_3$ (**2**)). Even if they display isomorph crystallographic structures, the magnetic studies revealed two different SCO behaviours without hysteretic effects: For complex **1**, the thermal variation of the $\chi_mT$ product showed an incomplete gradual HS–LS transition at $T_{1/2}$ = 205 K with residual fraction of the HS spin of *ca*. 10%., while complex **2** exhibits a complete and gradual HS to LS two-step-like transition centred around 245 K and 380 K. These magnetic data are in good agreement with the variable temperature infrared spectroscopy performed in the 2150–2200 cm$^{-1}$ range corresponding to the fundamental stretching vibration of NCS (**1**) and $NCBH_3$ (**2**) moieties. Furthermore, the thermal dependence of the intensity of the stretching vibration of NCS and $NCBH_3$ moieties (2150–220 cm$^{-1}$ in the case of complexes **1** and **2**) showed a clear correlation, including the intriguing two-step behaviour for complex **2**, with the thermal dependences of the $\chi_mT$ product derived from magnetic measurements for each complex. Finally, additional X-ray experiments at room temperature, where the second step of **2** is centred, revealed low diffraction power probably due to the low crystallinity of the sample, in agreement with the powder X-ray diffraction recorded on crushed single crystals that show a low crystallinity diffraction pattern.

**Supplementary Materials:** The following are available online at http://www.mdpi.com/2312-7481/6/2/26/s1, Scheme S1: Synthesis of tris(pyridin-2-yl)methanol (py$_3$C-OH), Scheme S2: Synthesis of tris(pyridin-2-yl)ethoxymethane (py$_3$C-OEt), Figure S1: $^1$H NMR (CDCl$_3$, 400 MHz, δ (ppm)) of py$_3$C-OH at 25 °C, Figure S2: $^{13}$C NMR (CDCl$_3$, 75 MHz, δ (ppm)) of py$_3$C-OH at 25 °C, Figure S3: IR spectrum of py$_3$C-OH at 25 °C, Figure S4: $^1$H NMR (CDCl3, 400 MHz, δ (ppm)) of py$_3$C-OEt at 25 C, Figure S5: $^{13}$C NMR (CDCl3, 125 MHz, δ (ppm)) of py$_3$C-OEt at 25°C, Figure S6: IR of py$_3$C-OEt at 25 °C, Figure S7: Molecular structure of **1-2** showing the asymmetric unit and the Fe(II) environments of the anionic ($[Fe(py_3C\text{-}OEt)(NCE)_3]^-$) and cationic ($[Fe(py_3C\text{-}OEt)_2]^{2+}$) moieties, Figure S8: Projection of the crystal packing in the *ac* plane (**a**) and π-π contact motif (**b**) of **1** (similar in **2**), Figure S9: 3D Hirshfeld surface maps and fingerprint plots of the intermolecular interactions around the $[Fe(py_3C\text{-}OEt)(NCE)_3]^-$ anions for compounds **1** and **2**, Figure S10: Experimental and simulated XRPD patterns for compounds **1** and **2**, Table S1: Crystal data and structural refinement parameters for complexes $[Fe(py_3C\text{-}OEt)_2][Fe(py_3C\text{-}OEt)(NCE)_3]_2{\cdot}2CH_3CN$ (E = S (**1**), BH$_3$ (**2**)), Table S2: Bond lengths and bond angles of compounds **1** and **2**.

**Author Contributions:** E.C. synthesized the ligand and the complexes with supervision of S.T.; and recorded and analyzed the infrared spectra with supervision of N.C. and S.P.; F.C. supervised the organic syntheses and interpreted the NMR spectra; E.C. analyzed the crystal data of the metal complexes with supervision of S.P.; S.B. performed the X-ray power diffraction and helped with the single crystal structures determination; C.J.G.-G. performed and interpreted the magnetic measurements; S.T. supervised the experimental work and wrote the manuscript on which all the authors have contributed. All authors have read and agreed to the published version of the manuscript.

**Funding:** This research was funded by the French CNRS (MITI Project), the "Université de Brest" (IBSAM institute), the Région Bretagne (EC), the Generalidad Valenciana (Prometeo2019/076 project) and the Spanish MINECO (Project CTQ2017-87201-P AEI/FEDER, EU).

**Conflicts of Interest:** The authors declare no conflict of interest.

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
