# Peer review of "Spin Cross-Over (SCO) Anionic Fe(II) Complexes Based on the Tripodal Ligand Tris(2-pyridyl)ethoxymethane"

_magnetochemistry, doi:10.3390/magnetochemistry6020026_

Round 1
Reviewer 1 Report
This is a nicely presented interesting contribution worth to be published. Experimental details and related interpretations are clearly presented and provide to new insights in spin-crossover species.
Author Response
Answer to Reviewer 1.
Comments. This is a nicely presented interesting contribution worth to be published. Experimental details and related interpretations are clearly presented and provide to new insights in spin-crossover species.
We thank this reviewer for his/her positive report
Reviewer 2 Report
Comments
The manuscript reports on syntheses, structures, infrared spectroscopic, and, magnetic properties of two iron(II) spin crossover complexes with the tripod-type ligand. For both compounds 1 and 2, each cationic and anionic mononuclear iron(II) unit is self-assembled through intermolecular hydrogen-bonds and p-p stacking interactions, and thereby is stabilized as ionic molecular hybrids in the crystal packing. Anionic mononuclear iron(II) units of compounds 1 and 2 show thermal spin crossover, which were investigated using variable temperature infrared spectroscopy and dc magnetic susceptibility in detail. The development of anionic spin crossover complexes is particularly important in the design of functional building units for multifunctional molecular solids. Therefore, I am sure that this manuscript will attract a lot of interest and many citations in the future. In addition, the manuscript is well organized and well written adequately. In conclusion, the manuscript is worth to be published in Magnetochemistry after minor changes and comments:
- I have the opinion that a view of crystal packing diagram of 1 should be inserted into the main text.
- If the authors can access to differential scanning calorimetry and variable temperature Mössbauer spectroscopy, I suggest to the authors to add measured data for 1 and 2. These data would give convincing arguments of thermal spin crossover and related phenomena of 1 and 2.
- Since the authors mention the thermochromism phenomenon of 1, I would appreciate the addition of photographs of single crystal of 1 at 293 K and 100 K in the supplementary material.
Author Response
Answer to Reviewer 2.
Comments. The manuscript reports on syntheses, structures, infrared spectroscopic, and, magnetic properties of two iron(II) spin crossover complexes with the tripod-type ligand. For both compounds 1 and 2, each cationic and anionic mononuclear iron(II) unit is self-assembled through intermolecular hydrogen-bonds and p-p stacking interactions, and thereby is stabilized as ionic molecular hybrids in the crystal packing. Anionic mononuclear iron(II) units of compounds 1 and 2 show thermal spin crossover, which were investigated using variable temperature infrared spectroscopy and dc magnetic susceptibility in detail. The development of anionic spin crossover complexes is particularly important in the design of functional building units for multifunctional molecular solids. Therefore, I am sure that this manuscript will attract a lot of interest and many citations in the future. In addition, the manuscript is well organized and well written adequately. In conclusion, the manuscript is worth to be published in Magnetochemistry after minor changes and comments:
1 - I have the opinion that a view of crystal packing diagram of 1 should be inserted into the main text.
Since the intermolecular contacts are not significant (in agreement with the gradual transition observed in both compounds), we have not inserted any crystal packing in the submitted version of manuscript, even if we have discussed different hydrogen bond contacts (C-H…S and C-H…B for 1 and 2, respectively) and the p-p interactions between the [Fe(py3C-OEt)(NCE)3]- anionic complexes, on the basis of 3D Hirshfeld surface maps and fingerprint plots (see Figure S9). However, we have added the crystal packing view (in the supplementary information) as suggested by the referee (see Figure S8).
2 - I the authors can access to differential scanning calorimetry and variable temperature Mössbauer spectroscopy, I suggest to the authors to add measured data for 1 and 2. These data would give convincing arguments of thermal spin crossover and related phenomena of 1 and 2.
We have access to DSC studies but since the transitions are gradual, we have not observed the expected endo and exothermic pics. For Mössbauer spectroscopy, we have external access from European collaborating group but unfortunately, the small quantities obtained for both samples would not allow correct studies.
3 - Since the authors mention the thermochromism phenomenon of 1, I would appreciate the addition of photographs of single crystal of 1 at 293 K and 100 K in the supplementary material.
The photographs of the single crystal used for X-ray data have been inserted in Table S1 (see revised SI)
Reviewer 3 Report
This paper by Triki et al describes spin crossover in a double salt iron(II) system. Emphasis is given to the anionic component. The paper is well written though long and wordy at times. The graphics are very good.
The authors should consider that they potentially had a 'double crossover' system, as described in a recent paper in Chem Comm 2019, 55, 14031 (Fe(II) cation; Fe(III) anion); this should be mentioned in the Introduction.
Emphasis needs to be given that spin crossover occurs only at Fe1 in the anionic moiety. A reason for why Fe2 remains LS should be given.
Experimentally, the paper lacks Mossbauer spectroscopy to explore Fe1 and Fe2 spin states; but the temperature dependent IR spectra are very good and not often given in spin crossover studies.
Are the authors sure that solvent is not (partially) lost when heating compound 2 to 500 K? Was tga measured?
Overall - a piece of good work acceptable to Magnetochemistry
Author Response
Answer to Reviewer 3.
Comments. This paper by Triki et al describes spin crossover in a double salt iron(II) system. Emphasis is given to the anionic component. The paper is well written though long and wordy at times. The graphics are very good.
1 - The authors should consider that they potentially had a 'double crossover' system, as described in a recent paper in Chem Comm 2019, 55, 14031 (Fe(II) cation; Fe(III) anion); this should be mentioned in the Introduction.
We are sorry for this oversight and we thank the referee for his/her help. We have modified the paragraph related to the anionic SCO complexes and we have added the references concerning the Fe(III) anionic complexes. See below the corresponding revised paragraph in the Introduction and the new references 18-23.
Modified paragraph (see revised introduction, all changes are indicted): “While, almost all the SCO materials, reported up today, are either cationic or neutral [1-17], the number of anionic complexes exhibiting the SCO phenomenon is relatively limited compared to the several hundreds of examples of neutral and cationic complexes reported every year. To the best of our knowledge, all the anionic SCO complexes are based on Fe(III) [18-23] or Fe(II) [24-30]. For those based on Fe(II), three different systems have been studied. The first system, reported in 2003 [24], concerns a series of Fe(II) complexes of chemical formula [FeIIH3LMe][FeIILMe]X (X- = ClO4-, BF4-, PF6-, AsF6-, SbF6-), based on the H3LMe (tris[2-(((2-methylimidazol-4-yl)methylidene)amino)ethyl]amine) acting as hexadente ligand in its neutral H3LMe form and with the deprotonated (LMe)3- anionic form to lead to a supramolecular 2D-array composed from [FeIIH3LMe]2+ cations and [FeIILMe]- anions, both exhibiting SCO. This system can be viewed as interacting anionic and cationic Fe(II) SCO species in a two-dimensional structure. The second example, reported more recently [25], concerns a trinuclear complex [FeII3(μ-L)6(H2O)6]6− based on the 4-(1,2,4-triazol-4-yl)ethanedisulfonate) (L) bridging anionic ligand, exhibiting a SCO transition above room temperature with a large hysteresis loop (> 85 K). The first example of the third system was reported in 2012 by T. Ishida et al. [26] as the first SCO example based on the rigid tetrakis(2-pyridyl)methane (py4C) ligand (see Scheme 1a, R = py). In this example, the Fe(II) mononuclear anionic complex, [Fe(py4C)(NCS)3], is surrounded by the py4C tripodal ligand acting as tridentate through the nitrogen atoms of three pyridyl groups and by three thiocyanate anions acting as terminal co-ligands (see Scheme 1b, R = py).”
- Phonsri, W.; Macedo, D.S.; Lewis, B. A. I.; Wain, D. F.; Murray, K. S. Iron(III) Azadiphenolate Compounds in a New Family of Spin Crossover Iron(II)–Iron(III) Mixed-Valent Complexes. Magnetochemistry 2019, 5, 37.
- Floquet, ; Boillot, M.-L.; Rivière, E.; Varret, F.; Boukheddaden, K.; Morineau, D.; Négrier, P. Spin transition with a large thermal hysteresis near room temperature in a water solvate of an iron(III) thiosemicarbazone complex. New J. Chem. 2003, 27, 341–348.
- Floquet, ; Guillou, N.; Négrier, P.; Rivière, E.; Boillot, M.-L. The crystallographic phase transition for a ferric thiosemicarbazone spin crossover complex studied by X-ray powder diffraction. New J. Chem. 2006, 30, 1621-1627.
- Cook, ; Habib, F.; Aharen, T.; Clérac, R.; Hu, A.; Murugesu, M. High-Temperature Spin Crossover Behavior in a Nitrogen-Rich FeIII Based System. Inorg. Chem. 2013, 52, 1825–1831.
- Takahashi, K.; Kawamukai, K.; Okai, M.; Mochida, T.; Sakurai, T.; Ohta, H.; Yamamoto, T.; Einaga, Y.; Shiota, Y.; Yoshizawa, K. A New Family of Anionic FeIII Spin Crossover Complexes Featuring a Weak-Field N2O4 Coordination Octahedron. Eur. J. 2016, 22, 1253-1257.
- Phonsri, ; Lewis, B. A. I.; Jameson, G. N. L.; Murray, K. S. Double spin crossovers: a new double salt strategy to improve magnetic and memory properties. Chem. Commun. 2019, 55, 14031-14034.
2 - Emphasis needs to be given that spin crossover occurs only at Fe1 in the anionic moiety. A reason for why Fe2 remains LS should be given.
The Fe2 is coordinated by two tripodal rigid and tridentate ligands leading to the [Fe(py3C-OEt)2]2+ cation complex. The reason for which the Fe2 remains LS is due essentially to the strong ligand field energy induced by the two tripodal ligands as observed in all parent complexes reported today (see ref. 26: Chem. Lett. 2012, 41, 716-718). In this case, it can be interesting to prepare similar anion using for example Co(II) ions which should be more appropriate for the design of complexes exhibiting SCO behaviour.
3 - Experimentally, the paper lacks Mossbauer spectroscopy to explore Fe1 and Fe2 spin states; but the temperature dependent IR spectra are very good and not often given in spin crossover studies.
Thank you for this interesting remark. Indeed, we have external access from European collaborating group for Mössbauer spectroscopy but unfortunately, the small quantities obtained for both samples would not allow correct studies.
4 - Are the authors sure that solvent is not (partially) lost when heating compound 2 to 500 K? Was tga measured?
This is an interesting question. As for compound 1, the magnetic properties of 2 were first measured in cooling mode (300-2 K) and then in warming mode (2-500 K), revealing almost similar magnetic behaviour. We agree with the referee that the measured sample above 355 K should correspond to the desolvated complex. However, the fact that the cooling and warming mode did not show significant difference in the magnetic behaviour (expected below 355 K), we can state that the solvent molecules do not affect significantly the SCO transition.
Overall - a piece of good work acceptable to Magnetochemistry.